# DISTANCE ESTIMATION FOR HIGH-DIMENSIONAL DISCRETE DISTRIBUTIONS

## ABSTRACT

We study the distance estimation problem for high-dimensional distributions. Given two distributions $\mathcal{P}$ and $\mathcal{Q}$ over $\{0,1\}^n$, and a parameter $\varepsilon$, the goal of distance estimation is to determine the statistical distance between the two distributions up to an additive tolerance $\pm\varepsilon$. Since exponential lower bounds (in $n$) are known for the problem in the standard sampling model, research has focused on models where one can draw conditional samples.

Among these models, *subcube conditioning* (SUBCOND), i.e., conditioning on arbitrary subcubes of the domain, holds the promise of widespread practical adoption owing to its ability to capture the natural behavior of distribution samplers. In this paper, we present the first polynomial sample distance estimator in the conditional sampling model, and our algorithm makes $\tilde{\mathcal{O}}(n^3/\varepsilon^5)$ SUBCOND queries.[1] We implement our algorithm to estimate the distance between distributions arising from real-life sampling benchmarks, and we find that our algorithm easily scales beyond the naive method.

## 1 INTRODUCTION

Given two discrete distributions $\mathcal{P}$ and $\mathcal{Q}$ over $\{0,1\}^n$, the total variation (TV) distance between $\mathcal{P}$ and $\mathcal{Q}$, denoted by $d_{TV}(P,Q)$, is defined as:

$$d_{TV}(\mathcal{P},\mathcal{Q}) = \frac{1}{2}\sum_{\sigma \in \{0,1\}^n} |\mathcal{P}(\sigma) - \mathcal{Q}(\sigma)|$$

In this paper, we are interested in the computation of $(\varepsilon,\delta)$-approximation of $d_{TV}(P,Q)$: i.e., we would like to compute an estimate $\kappa$ such that $\Pr[d_{TV}(\mathcal{P},\mathcal{Q}) - \varepsilon \leq \kappa \leq d_{TV}(\mathcal{P},\mathcal{Q}) + \varepsilon] \geq 1 - \delta$. TV distance is a fundamental notion in probability and finds applications in the diverse domains of computer science such as generative models (Goodfellow et al., 2014; Ji et al., 2023), MCMC algorithms (Andrieu et al., 2003; Boyd et al., 2004; Brooks et al., 2011), and probabilistic programming (Aguirre et al., 2021; Pote & Meel, 2022).

Theoretical investigations into the problem of TV distance computation have revealed the intractability of exact computation: In particular, the problem is #P-hard even when $\mathcal{P}$ and $\mathcal{Q}$ are represented as product distributions (Bhattacharyya et al., 2023a). Consequently, the focus has been on designing approximation techniques. When $\mathcal{P}$ and $\mathcal{Q}$ are specified explicitly, randomized polynomial time approximation schemes are known for some classes of distributions, such as Bayesian networks with bounded treewidth (Bhattacharyya et al., 2023b). Not every practical application allows explicit representation of probability distributions, and often, the output of some underlying process defines probability distributions. Accordingly, the field of distribution testing is concerned with the design of algorithmic techniques for different models of access to the underlying processes. Furthermore, in addition to the classical notion of time complexity, we are also concerned with the *query complexity*: how many queries do we make to a given access model?

The earliest investigations focused on the classical model of access where one is only allowed to access samples from $\mathcal{P}$ and $\mathcal{Q}$ (Paninski, 2008; Valiant & Valiant, 2011); however, a lower bound of $\Omega(2^n/n)$ (Valiant & Valiant, 2010; 2011) restricts the applicability of these estimators in practical scenarios. Consequently, there is a need to focus on more powerful models. In

---

[1] $\tilde{\mathcal{O}}$ will hide $\log(1/\varepsilon)$ factors.

this work, we will focus on the SUBCOND oracle access model owing to its ability to capture the behavior of probabilistic processes in several diverse settings (Jerrum et al., 1986; Chaudhuri et al., 1999; Zhao et al., 2018). Formally, SUBCOND oracle for a distribution $\mathcal{P}$ takes in a query setting $\rho \in \{0, 1, *\}^n$ and returns $\sigma \in S_\rho$ such that $\Pr[\sigma \text{ is output}] = \frac{\mathcal{P}(\sigma)}{\sum_{\pi \in S_\rho} \mathcal{P}(\pi)}$, wherein $S_\rho = \{\sigma \in \{0, 1\}^n | \forall i, (\rho_i = *) \lor (\sigma_i = \rho_i)\}$. It is worth remarking that while we use the name SUBCOND to be consistent with recent literature (Bhattacharyya & Chakraborty, 2018), there have been algorithmic frameworks since the late 1980s that have relied on the underlying query model (Jerrum et al., 1986).

The starting point of our investigation is the observation that, on the one hand, practical applications of distance estimation rely on heuristic methods and, accordingly, lack guarantees. On the other hand, no known algorithms, when given access to SUBCOND oracle, make less than $O(2^n/n)$ queries. The primary contribution of our work is to address the mentioned gap: we design the first algorithm that computes $(\varepsilon, \delta)$-approximation of TV distance and makes only polynomially many queries to SUBCOND oracle. Formally,

**Theorem 1.** *Given two distributions $\mathcal{P}$ and $\mathcal{Q}$ over $\{0, 1\}^n$, and the parameters $\varepsilon \in (0, 1)$, and $\delta \in (0, 1)$, the algorithm* DistEstimate$(\mathcal{P}, \mathcal{Q}, \varepsilon, \delta)$ *returns $\kappa$ such that*

$$\Pr[d_{TV}(\mathcal{P}, \mathcal{Q}) - \varepsilon \leq \kappa \leq d_{TV}(\mathcal{P}, \mathcal{Q}) + \varepsilon] \geq 1 - \delta$$

DistEstimate *makes $\tilde{\mathcal{O}}\left(n^3 \log(1/\delta)/\varepsilon^5\right)$ queries to the* SUBCOND *oracle.*

We now provide a high-level overview of DistEstimate: First observe $d_{TV}(P, Q) = \sum_\sigma \mathcal{Q}(\sigma) \cdot \max(1 - \mathcal{P}(\sigma)/\mathcal{Q}(\sigma), 0)$ and therefore, we can use the standard approach to sample $\sigma$ from $\mathcal{Q}$ and if we could determine $\mathcal{P}(\sigma)$ and $\mathcal{Q}(\sigma)$ up to some multiplicative factor, then we could set the value of the corresponding random variable to be $(1 - \mathcal{P}(\sigma)/\mathcal{Q}(\sigma), 0)$, and accordingly polynomially many samples would suffice to compute an approximation of $d_{TV}(P, Q)$. However, it is unlikely that it would be possible to approximate the value of $\mathcal{Q}(\sigma)$ for arbitrary $\sigma$ with only polynomially many queries to SUBCOND since $\mathcal{Q}(\sigma)$ can be arbitrarily small. The key technical contribution lies in showing that it suffices to ensure that with high probability, we can estimate $\mathcal{Q}(\sigma)$ and $\mathcal{P}(\sigma)$ with polynomially many queries to SUBCOND oracle where the probability space is defined over samples from $\mathcal{Q}$.

As mentioned earlier, our interest in the design of distance estimation techniques for the SUBCOND model stemmed from its ability to capture the behavior of probabilistic processes in practice. Therefore, we demonstrate the application of DistEstimate in a real-world setting. Sampling from discrete domains such as $\{0, 1\}^n$ under constraints is a hard problem; therefore, several heuristic-based samplers have been proposed over the years. We can view a sampler as a probabilistic process, and consequently, one is interested in measuring how far the distribution of a given sampler is from the ideal distribution. For our experiments we focus on samplers that sample satisfying assignments of a CNF, and SUBCOND is particularly well suited for this problem, as a CNF conditioned on a sub-cube is a CNF. We use a prototype of DistEstimate to evaluate the quality of two CNF samplers for different benchmarks. Our empirical evaluation demonstrates the promise of scalability: in particular, DistEstimate can handle benchmarks with $n = 70$ for which naive techniques would require $\simeq 10^{18}$ queries to samplers – a prohibitively large number.

**Organization** Section 3 defines the notation we use in most of the paper. We present the paper's main contribution, the estimator DistEstimate, along with its proof of correctness in Section 4. In Section 5, we present the result of the evaluation of our implementation of DistEstimate. Finally, we conclude in Section 6 and discuss some open problems. In the interest of exposition, we defer some proofs to the Appendix.

## 2 RELATED WORK

Distance estimation is one of the many problems in the broader area of distribution testing. Apart from estimation, there is extensive literature on the problems of identity and equivalence testing. The problem of identity testing involves returning Accept if $d_{TV}(\mathcal{P}, \mathcal{P}^*) = 0$ and returning Reject if $d_{TV}(\mathcal{P}, \mathcal{P}^*) > \varepsilon$, where $\mathcal{P}$ is an unknown distribution and $\mathcal{P}^*$ is known, i.e. you have a full description of $\mathcal{P}^*$. Equivalence testing is the generalization of identity testing. It is the problem of deciding between $d_{TV}(\mathcal{P}, \mathcal{Q}) = 0$ and $d_{TV}(\mathcal{P}, \mathcal{Q}) > \varepsilon$ where both $\mathcal{P}$ and $\mathcal{Q}$ are unknown. It is worth

emphasizing that for both identity and equivalence testing problems, any answer from the tester (Accept or Reject) is considered valid if $0 < d_{TV}(\mathcal{P}, \mathcal{Q}) \leq \varepsilon$. Provided only sample access, the sample complexity of identity testing is $\Theta\left(2^{n/2}/\varepsilon^2\right)$ (Paninski, 2008; Valiant & Valiant, 2017) and of equivalence testing is $\max(2^{2n/3}\varepsilon^{-4/3}, 2^{n/2}\varepsilon^{-2})$ (Chan et al., 2014; Valiant & Valiant, 2017). While testing is of theoretical interest, its practical application faces significant limitations primarily because testers must accept only when two given distributions are identical. In real-world scenarios, distributions are rarely identical but often exhibit close similarity. Consequently, a simplistic tester that consistently returns Reject can meet the specifications. A more rigorous definition of a tester is required to address this limitation, including estimating the distance between the two distributions. Unfortunately, this introduces a considerable challenge. Valiant & Valiant (2011) demonstrate that in the classical sampling model, the necessary number of queries increases to $2^n/n$, a significant jump from the previous $2^{2n/3}$

To sidestep the exponential lower bounds on testing, the conditional sampling model, or COND, was introduced independently by Chakraborty et al. and Canonne et al., and has been successfully applied to various problems, including identity and equivalence testing. In this model, the sample complexity of identity testing is $\Theta(\varepsilon^{-2})$ (independent of $n$), while for equivalence testing the best-known upper and lower bounds are $O(\log n/\varepsilon^{-5})$ (Falahatgar et al., 2015), and $\Omega(\sqrt{\log n})$ (Acharya et al., 2014) respectively. A survey by Canonne (2020) provides a detailed view of testing and related problems in various sampling models.

Our work investigates the distance estimation problem using the SUBCOND model, a restriction of COND. Unlike COND, which allows conditioning on arbitrary sets, the SUBCOND model allows conditioning only on sets that are subcubes of the domain. While COND significantly improves the sample complexity, it is not easily implementable in practice, as arbitrary subsets may not be efficiently represented and conditionally sampled. With a view towards plausible conditional models, Canonne et al. (2015); Bhattacharyya & Chakraborty (2018) came up with the SUBCOND model, which is particularly suited to the Boolean hypercube $\{0,1\}^n$. Canonne et al. (2021) used the SUBCOND model to construct a nearly-optimal $\Theta(\sqrt{n})$ uniformity testing algorithm for $\{0,1\}^n$, demonstrating its natural applicability for high-dimensional distributions. Then Chen et al. (2021) used SUBCOND to study the problems of learning and testing junta distributions supported on $\{0,1\}^n$. Bhattacharyya & Chakraborty (2018) developed a test for equivalence in the SUBCOND model, with query complexity of $O(n^2/\varepsilon^2)$. However, before this work, there was no distance estimation algorithm in the SUBCOND oracle model, and indeed even in the general COND model.

**Lower Bound** The problem of testing with SUBCOND access has a query complexity lower bound of $\Omega(n/\log(n))$ as a direct consequence of Theorem 11 of Canonne et al. (2020). For completeness, we formally state the lower bound and its proof in the appendix Section A.1.

## 3 NOTATIONS AND PRELIMINARIES

In this paper, we deal with discrete probability distributions over the $n$-dimensional discrete hypercube $\{0,1\}^n$. For any distribution $\mathcal{D}$ on $\{0,1\}^n$ and an element $\sigma \in \{0,1\}^n$, $\mathcal{D}(\sigma)$ is the probability of $\sigma$ in distribution $\mathcal{D}$. Similarly, for any set $S \subseteq \{0,1\}^n$, $\mathcal{D}(S)$ is the total probability of $S$ in $\mathcal{D}$. Further, $\sigma \sim \mathcal{D}$ represents that $\sigma$ is sampled from $\mathcal{D}$. The total variation (TV) distance of two probability distributions $\mathcal{P}$ and $\mathcal{Q}$ is defined as: $d_{TV}(\mathcal{P}, \mathcal{Q}) = \frac{1}{2} \sum_{\sigma \in \{0,1\}^n} |\mathcal{P}(\sigma) - \mathcal{Q}(\sigma)|$. For a random variable $v$, the expectation is denoted as $\mathbb{E}[v]$, and the variance as $\mathbb{V}[v]$. We use $[n]$ to represent the set $\{1, 2 \ldots, n\}$.

If $\sigma$ is a string of length $n > 0$, then $\sigma_i$ denotes the $i^{th}$ element of $\sigma$, and for $1 \leq j \leq n$, $\sigma_{<j}$ denotes the substring of $\sigma$ from 1 to $j-1$, $\sigma_{<j} = \sigma_1 \cdots \sigma_{j-1}$; similarly $\sigma_{\leq j} = \sigma_1 \cdots \sigma_j$, and $\sigma_{<1}$ denotes the empty (length 0) string, which we will also use $\phi$ to denote. For any string $\rho$ with $0 \leq |\rho| \leq n$, the subcube $S_\rho$ is defined as $S_\rho := \{w \in \{0,1\}^n | w_{\leq |\rho|} = \rho\}$

**Definition 1.** *A subcube conditioning oracle* SUBCOND$(\mathcal{D}, \rho)$ *takes as input a distribution $\mathcal{D}$ supported on $\{0,1\}^n$, and a query string $\rho$ with $0 \leq |\rho| \leq n$, and returns a sample $w \in \{0,1\}^n$ with probability $\mathcal{D}(w)/\mathcal{D}(S_\rho)$ if $w \in S_\rho$ and 0 otherwise.*

Since $S_\phi = \{0,1\}^n$, SUBCOND$(\mathcal{D}, \phi)$ returns a sample $w \in \{0,1\}^n$ as per the distribution $\mathcal{D}$. For any distribution $\mathcal{D}$, the distribution $\mathcal{D}^m_\rho$ denotes the marginal distribution of SUBCOND$(\mathcal{D}, \rho)$ in the $|\rho| + 1^{th}$ dimension, i.e. for $b \in \{0,1\}$, $\mathcal{D}^m_\rho(b) = \Pr_{w \sim \text{SUBCOND}(\mathcal{D},\rho)}[w_{|\rho|+1} = b]$

**Distance approximation**  We adapt the distance approximation algorithm of Bhattacharyya et al. (2020), that takes as input two distributions $\mathcal{P}$ and $\mathcal{Q}$, and provides an $(\theta, \delta)$ estimate of $d_{TV}(\mathcal{P}, \mathcal{Q})$. The algorithm has sample access to the two distributions and assumes the ability to approximately query the probability of any element $\sigma$ of the domain in $\mathcal{P}$ and $\mathcal{Q}$. Formally,

**Lemma 1.** *(Theorem 3.1 in (Bhattacharyya et al., 2020)) For $\delta, \theta \in (0, 1)$ and any distribution $\mathcal{P}$, $p_\sigma$ is called a $\theta$-estimate of $\mathcal{P}(\sigma)$, if $1 - \theta \leq p_\sigma/\mathcal{P}(\sigma) \leq 1 + \theta$. Then, for any two distributions $\mathcal{P}$ and $\mathcal{Q}$, given a set of samples $S$ from $\mathcal{P}$, along with the $\theta$-estimates $p_\sigma$ and $q_\sigma$ for each $\sigma \in S$, let $Z$ be the estimate $Z = \frac{1}{|S|} \sum_{i \in S} 1_{q_\sigma > p_\sigma} \left(1 - \frac{p_\sigma}{q_\sigma}\right)$. If $|S| \geq \frac{(1-\theta)\log(2/\delta)}{8\theta^2}$, then $d_{TV}(\mathcal{P}, \mathcal{Q}) + 4\theta/(1-\theta) \leq Z \leq d_{TV}(\mathcal{P}, \mathcal{Q}) + 4\theta/(1-\theta)$ with probability $1 - \delta$.*

We defer the proof to appendix Section A.2.

### 3.1 Taming Distributions

To estimate the probability of an element $\sigma$ in a distribution $\mathcal{D}$ using the SUBCOND oracle, the query complexity will turn out to be $\Omega(1/\mathcal{D}^m_{\sigma_{<\ell}}(\sigma_\ell))$. The query complexity can hence be arbitrarily high since we don't have any lower bound on the marginal $\mathcal{D}^m_{\sigma_{<\ell}}(\sigma_\ell)$. To bound the complexity, we show a distribution $\mathcal{D}'$ defined over $\{0,1\}^n$ that is close to $\mathcal{D}$ and has the property that all of its marginals are not too small. To this end, we adapt the $\theta$-balancing trick, devised for product distributions in (Canonne et al., 2020, Thm. 6), to show that for $\theta \in (0, 1/2)$ we can modify $\mathcal{D}$ to get $\mathcal{D}'$ such that $\mathcal{D}'^m_{\sigma_{<\ell}}(\sigma_\ell) \in [\theta, 1 - \theta]$, for every $\ell$ and every $\sigma$. Thus we can avoid the problem of very small marginals by simulating access to a different but close distribution.

**Definition 2.** *A distribution $\mathcal{D}'$ is called $\theta$-tamed if for all $\sigma \in \{0,1\}^n$ and for all $\ell \in [n]$, we have $\mathcal{D}'^m_{\sigma_{<\ell}}(\sigma_\ell) \in [\theta, 1 - \theta]$.*

Given any distribution $\mathcal{D}$, we will now show there exists a $\theta$-tamed distribution $\mathcal{D}'$ such that $d_{TV}(\mathcal{D}, \mathcal{D}') \leq \theta n$. Furthermore given SUBCOND access to $\mathcal{D}$ we can also make SUBCOND queries on the distribution $\mathcal{D}'$. First, we present a randomized process that generates $\sigma \sim \mathcal{D}'$.

Given a distribution $\mathcal{D}$ and $\theta \in (0, 1/2)$, consider a randomized procedure that generates an element $\sigma \sim \{0,1\}^n$ as follows: for all $i \geq 1$, having generated the substring $\sigma_{<i}$, set $\sigma_i = 0$ with probability $(1-2\theta)\mathcal{D}^m_{\sigma_{<i}}(0)+\theta$ and $\sigma_i = 1$ with probability $(1-2\theta)\mathcal{D}^m_{\sigma_{<i}}(1)+\theta$. The distribution corresponding to the above randomized procedure is the distribution $\mathcal{D}'$.

Note that for all $\ell \in [n]$, $c \in \{0,1\}$, and $\rho \in \{0,1\}^{\ell-1}$, we have $\mathcal{D}'^m_\rho(c) = (1 - 2\theta)\mathcal{D}^m_\rho(c) + \theta$. Thus to implement the SUBCOND query $\mathcal{D}'^m_\rho$, with probability $1 - 2\theta$ return the result of $\mathcal{D}^m_\rho$, or else with probability $2\theta$ draw a sample uniformly from $\{0,1\}$.

**Lemma 2.** *For $\theta \in (0, 1/2)$, distribution $\mathcal{D}$, and its $\theta$-tamed version $\mathcal{D}'$, we have $d_{TV}(\mathcal{D}, \mathcal{D}') \leq \theta n$.*

The proof of Lemma 2 is deferred to the appendix Section C.

## 4 DistEstimate: an algorithm to estimate the distance between distributions using SUBCOND

We first present the pseudocode of our algorithm DistEstimate, and the SubToEval and DistEstimateCore subroutines.

**Algorithm 1:** DistEstimate($\mathcal{P}, \mathcal{Q}, \varepsilon, \delta$)

1   $K \leftarrow 48 \log(1/\delta)$
2   $\forall_{j \in [K]} r_j \leftarrow 0$
3   **forall** $j \in [K]$ **do**
4     |   $r_j \leftarrow$ DistEstimateCore($\mathcal{P}, \mathcal{Q}, \varepsilon$)
5   **return** $\texttt{Median}_j(r_j)$

**Algorithm 2:** SubToEval($\mathcal{D}, \varepsilon, \sigma$)

1   $k \leftarrow \lceil 4n/\varepsilon^2 \rceil$
2   $thr \leftarrow 15 \lceil 8n^2/\varepsilon^2 \rceil$
3   $C \leftarrow 0$
4   $\forall_{j \in [n]} x_j \leftarrow 0$
5   **forall** $j \in [n]$ **do**
6     |   $t \leftarrow 0$
7     |   **while** $t < k$ **do**
8     |     |   $C \leftarrow C + 1$
9     |     |   $x_j \leftarrow x_j + 1$
10     |     |   $\alpha \leftarrow$ SUBCOND($\mathcal{D}, \sigma_{<j}$)
11     |     |   **if** $C > thr$ **then return** $0$
12     |     |   **if** $\alpha_j = \sigma_j$ **then** $t \leftarrow t + 1$
13   **return** $\prod_{j=1}^{n} k/x_j$

**Algorithm 3:** DistEstimateCore($\mathcal{P}, \mathcal{Q}, \varepsilon$)

1   $\theta \leftarrow \varepsilon/(\varepsilon + 4\frac{4}{9})$
2   $m \leftarrow (1 - \theta) \log(80/3)/8\theta^2$
3   $T \leftarrow 48 \log(4m)$
4   $S \leftarrow [0] * m$
5   $\mathcal{P}' \leftarrow \texttt{Tame}(\varepsilon/10n, \mathcal{P})$
6   **forall** $i = 1$ **to** $m$ **do**
7     |   $\sigma \leftarrow$ SUBCOND($\mathcal{Q}, \phi$)
8     |   **forall** $j = 1$ **to** $T$ **do**
9     |     |   $p_{ij} \leftarrow$ SubToEval($\mathcal{P}', \theta, \sigma$)
10     |     |   $q_{ij} \leftarrow$ SubToEval($\mathcal{Q}, \theta, \sigma$)
11     |   $p_i \leftarrow \texttt{Median}_j(p_{ij})$
12     |   $q_i \leftarrow \texttt{Median}_j(q_{ij})$
13     |   **if** $q_i > p_i$ **then**
14     |     |   $S[i] \leftarrow 1 - p_i/q_i$
15   **return** $\sum_{i \in [m]} S[i]/m$

We will now give a high-level overview of the two algorithms, followed by a formal analysis.

#### 4.1.1 OUTLINE OF THE DistEstimate AND DistEstimateCore ROUTINES

The pseudocode of DistEstimate and DistEstimateCore is given in Algorithm 1 and 3 respectively. DistEstimate takes as input two distributions $\mathcal{P}$ and $\mathcal{Q}$ defined over the support $\{0, 1\}^n$, along with the parameter $\varepsilon$ for tolerance and the parameter $\delta$ for confidence, and returns an $\varepsilon$-approximate estimate of $d_{TV}(\mathcal{P}, \mathcal{Q})$ with probability at least $1 - \delta$. The DistEstimateCore subroutine call returns an $\varepsilon$-approximate estimate of $d_{TV}(\mathcal{P}, \mathcal{Q})$ with probability at least $5/6$, and DistEstimate calls the DistEstimateCore subroutine $K = \log(1/\delta)$ times to boost the overall probability to $1 - \delta$ with the help of the median trick.

DistEstimateCore starts by computing the constants $\theta$, $m$, and $T$, where $m$ and $T$ are the numbers of iterations of the outer and inner loop, respectively, and they solely depend on $\varepsilon$. DistEstimateCore then calls the subroutine $\texttt{Tame}$(Line 5) on the input $\mathcal{P}$ to simulate another distribution $\mathcal{P}'$, that is $\varepsilon/10$ close to $\mathcal{P}$ in TV distance, and has the property that all of the marginal probabilities are lower bounded by $\Omega(1/n)$. This property of $\mathcal{P}'$ is crucial for the polynomial query complexity of DistEstimateCore. The construction of $\mathcal{P}'$, and claimed guarantee, is discussed in Section 3.1.

DistEstimateCore then draws $m$ samples $\sigma \sim \mathcal{Q}$, and for each sample $\sigma$, calls SubToEval $T$ times to find the $(1 \pm \theta)$ estimates of $\mathcal{Q}(\sigma)$ and $\mathcal{P}'(\sigma)$. The value of $T$ is chosen to be high enough to ensure that the medians of the estimates, $p_i$ and $q_i$, are correct with the required confidence. DistEstimateCore then computes the distance using these estimates as given in Lemma 1.

#### 4.1.2 OUTLINE OF THE SubToEval SUBROUTINE

The SubToEval subroutine takes as input an element $\sigma \in \{0, 1\}^n$, a distribution $\mathcal{D}$ over $\{0, 1\}^n$ and a parameter $\varepsilon$, and it provides $\pm \varepsilon$ estimate of $\mathcal{D}(\sigma)$ with high probability.

**Lemma 3.** *The subroutine* SubToEval($\mathcal{D}, \varepsilon, \sigma$) *takes as input a distribution $\mathcal{D}$ over $\{0, 1\}^n$, an element $\sigma \in \{0, 1\}^n$, and $\varepsilon \in (0, 1/2)$ and makes at most $15\lceil 8n^2/\varepsilon^2 \rceil$ calls to* SUBCOND *oracle. It returns $\widehat{\mathcal{D}}(\sigma)$, such that $\Pr[\widehat{\mathcal{D}}(\sigma) \in (1 \pm \varepsilon)\mathcal{D}(\sigma)] \geq 3/5$.*

The proof of the aforementioned lemma will be detailed in Section 4.2.1. This section provides a concise description and an informal discussion of the algorithm. The probability $\mathcal{D}(\sigma)$ can be expressed as a product of marginals, $\mathcal{D}(\sigma) = \prod_{i=1}^{n} \mathcal{D}_{\sigma_{<i}}^{m}(\sigma)$, by applying the chain rule of probability.

Essentially, the subroutine approximates each marginal $\mathcal{D}^m_{\sigma_{<j}}(\sigma)$ by $k/x_j$ for each $j \in [n]$, utilizing the SUBCOND oracle. The product $\prod_{j=1}^n k/x_j$ is then employed as the final estimate for $\mathcal{D}(\sigma)$.

In this context, the variable $x_j$ represents the total count of SUBCOND$(\mathcal{D}, \sigma_{<j})$ queries executed until $k$ occurrences of $\sigma_j$ are observed. Given that $\mathcal{D}^m_\rho(b) = \Pr_{w \sim \text{SUBCOND}(\mathcal{D},\rho)}[w_{|\rho|+1} = b]$ for any $\rho$ (as discussed in Section 3), the ratio $k/x_j$ is an intuitive choice as an estimator for $\mathcal{D}^m_{\sigma_{<j}}(\sigma)$. Moreover, the subroutine monitors the total number of calls to the SUBCOND oracle in the variable $C$. If this count exceeds the threshold $thr = 15\lceil 8n^2/\varepsilon^2 \rceil$ at any point, the subroutine terminates and returns 0.

To estimate $\mathcal{D}(\sigma)$, it is essential to estimate each of the $n$ marginals, $\mathcal{D}^m_{\sigma_{<i}}(\sigma)$, to within an error margin of approximately $1 + \varepsilon/n$. Naively, this would require at least $n^2/\mathcal{D}^m_{\sigma_{<i}}(\sigma)$ queries for each marginal, not accounting for the dependence on $\varepsilon$. Consequently, the total expected query complexity would sum up to $\sum_{i=1}^n n^2/\mathcal{D}^m_{\sigma_{<i}}(\sigma)$. This quantity is at least $\Omega(n^3)$, but it could potentially be unbounded as $\mathcal{D}^m_{\sigma_{<i}}(\sigma)$ can take arbitrary values. In the forthcoming section, we reduce this complexity to $O(n^2)$ through a more nuanced analysis.

### 4.2.1 ANALYSIS OF SubToEval

In this section, we prove Lemma 3 which is the main technical content of the paper. To prove Lemma 3, we consider the subroutine $\text{SubToEval}'(\mathcal{D}, \varepsilon, \sigma)$ as follows:

---

**Algorithm 4:** $\text{SubToEval}'(\mathcal{D}, \varepsilon, \sigma)$

---
1   $k \leftarrow \lceil 4n/\varepsilon^2 \rceil$
2   $\forall_{j \in [n]} x_j \leftarrow 0$
3   **forall** $j \in [n]$ **do**
4     $t \leftarrow 0$
5     **while** $t < k$ **do**
6       $x_j \leftarrow x_j + 1$
7       $\alpha \leftarrow \text{SUBCOND}(\mathcal{D}, \sigma_{<j})$
8       **if** $\alpha_j = \sigma_j$ **then** $t \leftarrow t + 1$
9   **return** $\prod_{j=1}^n k/x_j$

---

We now differentiate between the subroutines $\text{SubToEval}(\mathcal{D}, \varepsilon, \sigma)$ and $\text{SubToEval}'(\mathcal{D}, \varepsilon, \sigma)$. The behavior of $\text{SubToEval}'(\mathcal{D}, \varepsilon, \sigma)$ mirrors that of $\text{SubToEval}(\mathcal{D}, \varepsilon, \sigma)$ except in one key aspect: in $\text{SubToEval}(\mathcal{D}, \varepsilon, \sigma)$, when the number of calls to the SUBCOND oracle surpasses a certain threshold $15\lceil 8n^2/\varepsilon^2 \rceil$, the subroutine immediately returns 0 and terminates. In contrast, $\text{SubToEval}'(\mathcal{D}, \varepsilon, \sigma)$ does not impose such a limit, allowing an unlimited number of calls to the SUBCOND oracle. This modification is critical for our analysis because it results in the variable $x_j$ in $\text{SubToEval}'(\mathcal{D}, \varepsilon, \sigma)$ following the well-known negative binomial distribution.

In Lemma 5, we demonstrate that $\text{SubToEval}'(\mathcal{D}, \varepsilon, \sigma)$ correctly returns the $\pm\varepsilon$ estimate of $\mathcal{D}(\sigma)$ with high probability. Following this, Lemma 6 establishes an upper bound on the expected number of calls to the SUBCOND oracle within the subroutine. Concluding this section, we present the detailed proof of our main Lemma 3.

We now turn our attention to proving Lemma 5. Consider a discrete r.v. that takes the value $v$ with probability $p$. The count of trials required to observe $r$ instances of $v$ follows a negative binomial distribution, denoted as $\text{NB}(r, p)$. It is important to note that the expected value $\mathbb{E}[\text{NB}(r, p)]$ is $r/p$, and its variance $\mathbb{V}[\text{NB}(r, p)]$ is $r(1 - p)/p^2$. In our case, the r.v. $x_j$ in Algorithm 4 follows the distribution $\text{NB}(k, \mathcal{D}^m_{\sigma_{<j}}(\sigma_j))$. This is explicitly designed in lines 5-8 of Algorithm 4. The rationale is that in $\text{SubToEval}'$, $x_j$ is a discrete r.v. representing the count of SUBCOND queries made to $\mathcal{D}^m_{\sigma_{<j}}$ before $k$ occurrences of $\sigma_j$ are encountered. We will formalize this observation in the upcoming lemma, the proof of which we relegate to the appendix section D.

**Lemma 4.** *For $j \in [n]$, the random variable $x_j$ (in Algorithm 4) is distributed as* $\text{NB}(k, \mathcal{D}^m_{\sigma_{<j}}(\sigma_j))$

Note that $\mathcal{D}(\sigma) = \prod_{i=1}^n \mathcal{D}^m_{\sigma_{<i}}(\sigma)$. Our estimator for the marginal $\mathcal{D}^m_{\sigma_{<j}}(\sigma_j)$ is $k/x_j$, and $x_j$ is distributed as $\text{NB}(k, \mathcal{D}^m_{\sigma_{<j}}(\sigma_j))$.

**Lemma 5.** *The subroutine* $\mathsf{SubToEval}'(\mathcal{D}, \varepsilon, \sigma)$ *takes as input a distribution $\mathcal{D}$ over $\{0,1\}^n$, an element $\sigma \in \{0,1\}^n$, and $\varepsilon \in (0, 1/2)$. It returns $\widehat{\mathcal{D}}(\sigma)$, such that $\Pr[\widehat{\mathcal{D}}(\sigma) \in (1 \pm \varepsilon)\mathcal{D}(\sigma)] \geq 2/3$.*

*Proof.* We use Chebyshev's bound to prove the lemma, using a variance reduction technique introduced by Dyer & Frieze (1991). We start by defining a random variable $Z = \prod_{j=1}^n k/x_j$, where $x_j$ and $k$ refer to their values in Algorithm 4. Since $\mathcal{D}(\sigma) = \prod_{j=1}^n \mathcal{D}^m_{\sigma_{<j}}(\sigma_j)$, and $k/x_j$ is an estimator for $\mathcal{D}^m_{\sigma_{<j}}(\sigma_j)$, the product ($\prod_{j=1}^n k/x_j$) is the estimator for $\mathcal{D}(\sigma)$.

Furthermore,

$$\frac{\mathbb{V}[\prod_{j=1}^n x_j/k]}{\mathbb{E}[\prod_{i=1}^n x_j/k]^2} = \frac{\mathbb{E}[(\prod_{j=1}^n x_j/k)^2]}{\mathbb{E}[\prod_{i=1}^n x_j/k]^2} - 1 = \prod_{j=1}^n \frac{\mathbb{E}[(x_j/k)^2]}{\mathbb{E}[x_j/k]^2} - 1 = \prod_{j=1}^n \left(1 + \frac{\mathbb{V}[x_j/k]}{\mathbb{E}[x_j/k]^2}\right) - 1$$

Substituting, $\mathbb{V}[x_j/k]$ and $\mathbb{E}[x_j/k]^2$, we have

$$\frac{\mathbb{V}[\prod_{j=1}^n x_j/k]}{\mathbb{E}[\prod_{i=1}^n x_j/k]^2} = \prod_{j=1}^n \left(1 + \frac{k(1-p)/(p^2 k^2)}{(k/p)^2/k^2}\right) - 1 = \prod_{j=1}^n \left(1 + \frac{1-p}{k}\right) - 1 \leq \prod_{j=1}^n \left(1 + \frac{1}{k}\right) - 1$$

Substituting value of $k$, we have

$$\frac{\mathbb{V}[\prod_{j=1}^n x_j/k]}{\mathbb{E}[\prod_{i=1}^n x_j/k]^2} \leq \left(1 + \frac{\varepsilon^2}{4n}\right)^n - 1 \leq \exp\left(\frac{\varepsilon^2}{4}\right) - 1 \leq \frac{\varepsilon^2}{3}$$

The last inequality comes from the fact that for $x \in (0,1), k > 1, \exp\left(\frac{x}{k+1}\right) \leq 1 + \frac{x}{k}$. Then, from Chebyshev's inequality,

$$\Pr\left[|\mathbb{E}[Z] - Z| \geq \mathbb{E}[Z]\varepsilon\right] \leq \mathbb{V}[Z]/\varepsilon^2 \mathbb{E}[Z]^2 \leq 1/3$$

Since $Z$ was the estimate $\widehat{\mathcal{D}}(\sigma)$, we have shown that $\Pr[\widehat{\mathcal{D}}(\sigma) \in (1 \pm \varepsilon)\mathcal{D}(\sigma)] \geq 2/3$. $\qquad\square$

We will now show an upper bound on the query complexity of $\mathsf{SubToEval}'$. The number of SUBCOND queries made by $\mathsf{SubToEval}'$ is a distribution that depends upon the input $\mathcal{D}, \varepsilon$, and $\sigma$. We define a random variable $\mathsf{QC}(\mathsf{SubToEval}'(\mathcal{D}, \varepsilon, \sigma))$, that represents the query complexity of the $\mathsf{SubToEval}'$ routine. The following lemma asserts that when $\sigma \sim \mathcal{D}$, the query complexity of $\mathsf{SubToEval}'$ with input $\mathcal{D}$ is in expectation $\lceil 8n^2/\varepsilon^2 \rceil$.

**Lemma 6.** *For a distribution $\mathcal{D}$ defined over $\{0,1\}^n$, $\varepsilon \in (0,1)$,*

$$\mathbb{E}_{\sigma \sim \mathcal{D}}\left[\mathbb{E}\left[\mathsf{QC}(\mathsf{SubToEval}'(\mathcal{D}, \varepsilon, \sigma))\right]\right] = \lceil 8n^2 \varepsilon^{-2} \rceil$$

*where the inner expectation is over the internal randomness of* $\mathsf{SubToEval}'$

We defer the proof to the appendix Section E.

*Proof of Lemma 3.* From Lemma 5, we have that the expected number of queries of the subroutine $\mathsf{SubToEval}'(\mathcal{D}, \varepsilon, \sigma)$ is $\lceil 8n^2 \varepsilon^{-2} \rceil$. Therefore, by Markov's inequality, the probability that the subroutine $\mathsf{SubToEval}'(\mathcal{D}, \varepsilon, \sigma)$ makes $15\lceil 8n^2 \varepsilon^{-2} \rceil$ queries is at most $1/15$. Further, from Lemma 5, the returned value $\widehat{\mathcal{D}}(\sigma)$ satisfies $\widehat{\mathcal{D}}(\sigma) \in (1 \pm \varepsilon)\mathcal{D}(\sigma)$ with probability $2/3$. Thus by union bound, with probability at least $2/3 - 1/15 \geq 3/5$, the $\mathsf{SubToEval}'(\mathcal{D}, \varepsilon, \sigma)$ makes at most $15\lceil 8n^2 \varepsilon^{-2} \rceil$ queries and additionally, the returned value $\widehat{\mathcal{D}}(\sigma)$ satisfies $\widehat{\mathcal{D}}(\sigma) \in (1 \pm \varepsilon)\mathcal{D}(\sigma)$. This implies that the subroutine $\mathsf{SubToEval}(\mathcal{D}, \varepsilon, \sigma)$ returns $\widehat{\mathcal{D}}(\sigma)$, such that $\Pr[\widehat{\mathcal{D}}(\sigma) \in (1 \pm \varepsilon)\mathcal{D}(\sigma)] \geq 3/5$. $\qquad\square$

### 4.2.2 ANALYSIS FOR DistEstimateCore AND DistEstimate

**Theorem 2.** *Given two distributions $\mathcal{P}$ and $\mathcal{Q}$ over $\{0,1\}^n$, and the parameters $\varepsilon \in (0,1)$, the algorithm* $\mathsf{DistEstimateCore}(\mathcal{P}, \mathcal{Q}, \varepsilon)$ *returns $\kappa$ such that*

$$\Pr[d_{TV}(\mathcal{P}, \mathcal{Q}) - \varepsilon \leq \kappa \leq d_{TV}(\mathcal{P}, \mathcal{Q}) + \varepsilon] \geq 5/6$$

$\mathsf{DistEstimateCore}$ *makes $\tilde{\mathcal{O}}\left(n^3/\varepsilon^5\right)$ queries to the* SUBCOND *oracle.*

*Proof.* Let Good be the event that DistEstimateCore returns an approximately correct $\kappa$, i.e. $d_{TV}(\mathcal{P}, \mathcal{Q}) - \varepsilon \leq \kappa \leq d_{TV}(\mathcal{P}, \mathcal{Q}) + \varepsilon$. We want to show that $\Pr[\text{Good}] \geq 5/6$.

Recall that $\mathcal{P}'$ (Line 5 of Alg. 3) is a tamed version of $\mathcal{P}$. Then for all $i \in [m]$, let $\text{Bad}_i^p$ be the event that in the $i^{th}$ iteration of the inner loop (Line 11) of DistEstimateCore, $p_i \notin (1 \pm \theta)\mathcal{P}'(\sigma)$, and similarly let $\text{Bad}_i^q$ be the event that $q_i \notin (1 \pm \theta)\mathcal{Q}(\sigma)$. Then let $\text{Bad} = \bigcup_{i \in [m]}(\text{Bad}_i^p \cup \text{Bad}_i^q)$, be the event that at least one of the estimates is incorrect.

From the value of $T$(Line 3), and our choice of approximation parameter $\theta$ (on Lines 9,10), and from Lemma 3 we know that in iteration $j$ of the inner loop, $\Pr[p_{ij} \in \mathcal{P}'(\sigma)(1 \pm \theta)] \geq 2/3$ and also that $\Pr[q_{ij} \in \mathcal{Q}(\sigma)(1 \pm \theta)] \geq 2/3$. We use the median trick to find the higher confidence estimates $p_i \leftarrow \text{Median}_j(p_{ij})$ and $q_i \leftarrow \text{Median}_j(q_{ij})$, such that $\forall i \in [m], \Pr[\text{Bad}_i^p], \Pr[\text{Bad}_i^q] \leq 1/12m$. Then,

$$\Pr[\text{Bad}] = \Pr\left[\bigcup_{i \in [m]}(\text{Bad}_i^p \cup \text{Bad}_i^q)\right] \leq \sum_{i \in [m]}(\Pr[\text{Bad}_i^p] + \Pr[\text{Bad}_i^q]) = \frac{1}{12}$$

From Lemma 1, setting $\theta = \varepsilon/(40/9 + \varepsilon)$, and the choice of $m$ (Line 2) tell us that the estimate $\sum_{i \in [m]} S[i]/m$ (on Line 15 of Alg. 3) is a $\pm 9\varepsilon/10$ estimate of $d_{TV}(\mathcal{P}', \mathcal{Q})$, with probability at least $1 - 1/12$ (assuming $\overline{\text{Bad}}$). We denote the estimate as $\kappa$, then $\kappa - 9\varepsilon/10 \leq d_{TV}(\mathcal{P}', \mathcal{Q}) \leq \kappa + 9\varepsilon/10$. Furthermore, since $\mathcal{P}'$ is a $\varepsilon/10n$-tamed version of $\mathcal{P}$ (Line 5 of Alg 3), from Lemma 2 we know that $d_{TV}(\mathcal{P}', \mathcal{P}) \leq \varepsilon/10$. Then, from the triangle inequality, we have two bounds on $d_{TV}(\mathcal{P}, \mathcal{Q})$:

$$d_{TV}(\mathcal{P}, \mathcal{Q}) \leq d_{TV}(\mathcal{P}', \mathcal{Q}) + d_{TV}(\mathcal{P}', \mathcal{P}) \leq \kappa + 9\varepsilon/10 + \varepsilon/10 = \kappa + \varepsilon$$
$$d_{TV}(\mathcal{P}, \mathcal{Q}) \geq d_{TV}(\mathcal{P}', \mathcal{Q}) - d_{TV}(\mathcal{P}', \mathcal{P}) \geq \kappa - 9\varepsilon/10 - \varepsilon/10 = \kappa - \varepsilon$$

Thus, from the union bound, we have that $\Pr[\text{Good}] \geq 1 - 1/12 - \Pr[\text{Bad}] = 1 - 2/12 = 5/6$. $\quad\square$

Using the standard median trick, DistEstimate boosts the success probability from $5/6$ (in DistEstimateCore) to $1 - \delta$. We show this in the proof of Theorem 1 in the appendix Section B.

## 5 APPLICATION: MEASURING THE QUALITY OF SAMPLERS

We demonstrate the practical utility of DistEstimateCore by focusing on one use case: evaluating the quality of constrained samplers. Constrained samplers are fundamental objects in the field of constraint solving. In particular, a sampler takes in a set of constraints, which can be viewed as representing a function $f : \{0,1\}^n \rightarrow \{0,1\}$, and we are interested in drawing uniform samples from the set $\{\sigma \mid f(\sigma) = 1\}$: another interpretation is that we are interested in sampling from the set of all the assignments that satisfy the given set of constraints. It is worth remarking that even determining whether the set $\{\sigma \mid f(\sigma) = 1\}$ is non-empty is the prototypical NP-complete problem. Therefore, it is often the case that constrained samplers that can scale to large instances are often complex heuristic-based systems that are not amenable to theoretical analysis. Accordingly, there is a significant interest in designing techniques that can evaluate the quality of such samplers. We will focus on two state-of-the-art samplers: STS (Ermon et al., 2012) and CMSGen (Soos et al., 2020). We seek to evaluate the quality of these samplers with respect to a reference sampler WAPS, which is an exact sampler, i.e., provides rigorous theoretical guarantees of uniformity; however, such guarantees are attained at the cost of scalability. It is worth remarking that SUBCOND is a natural query model in the context of constrained samplers, as every query string can be represented as a conjunction of constraints.

We choose the experiment's tolerance parameter to be $\varepsilon = 0.5$. The experiments were conducted on a cluster with AMD EPYC 7713 CPU cores. For each benchmark, we use 32 cores with 4GB of memory. We evaluate our implementation on two datasets comprising Boolean formulas: (1) scalable and (2) real-world. The scalable benchmarks consist of random 3-CNFs defined over $n$ variables where $n \in \{30, 35, \ldots, 70\}$. The real-world instances were drawn from a publicly available set of circuits that have been used in the benchmarking of and samplers (Meel, 2020).

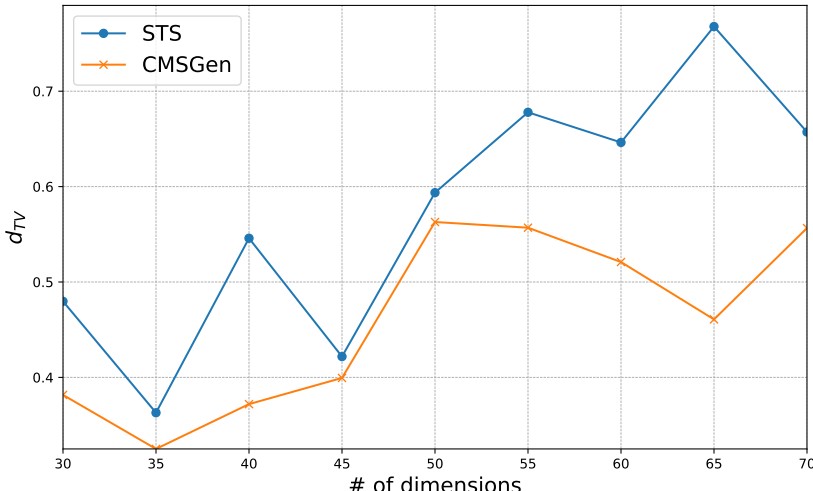

Figure 1: TV Distance as returned by DistEstimateCore, for STS and CMSGen.

**Results** We find that DistEstimateCore terminated with a result on all benchmarks. We present the results of our experiments on real-world instances in Table 1. The first column indicates the benchmark's name, the second the number of dimensions of the support, and the following four columns display the number of samples and the time required for the test for STS and CMSGen, respectively.

We present the TV distance, as computed by DistEstimateCore, in Figure 1. We use scalable instances for this experiment. As seen in the graph, the distance of both CMSGen and STS, from WAPS, increases with the number of dimensions.

Table 1: The sample complexity and runtime performance of DistEstimateCore on 7 real-world instances.

| Benchmark | Dimensions | STS | | CMSGen | |
|---|---|---|---|---|---|
| | | # of samples | time (in s) | # of samples | time (in s) |
| s1196a_3_2 | 33 | 1.8e+09 | 4.1e+05 | 1.9e+09 | 5.3e+05 |
| 53.sk_4_32 | 33 | 1.7e+09 | 2.5e+05 | 1.9e+09 | 1.6e+06 |
| s1238a_3_2 | 33 | 1.8e+09 | 4.0e+05 | 1.9e+09 | 5.5e+05 |
| 27.sk_3_32 | 33 | 1.7e+09 | 1.9e+05 | 1.9e+09 | 1.0e+06 |
| s1196a_7_4 | 33 | 1.8e+09 | 4.6e+05 | 1.9e+09 | 5.5e+05 |
| s420_15_7 | 35 | 2.1e+09 | 4.2e+05 | 2.3e+09 | 4.0e+05 |
| 111.sk_2_36 | 37 | 2.2e+09 | 3.5e+05 | 8.3e+08 | 6.6e+05 |

## 6 CONCLUSION

This paper focused on the distance estimation problem in the SUBCOND model. We sought to alleviate the two major weaknesses of the prior state of the art: the estimators required a prohibitively large number of queries, and they could only test equivalence in polynomially many queries. Our primary contribution, DistEstimateCore, enables distance estimation and requires only $\mathcal{O}(n^3/\varepsilon^5)$ queries. An interesting direction of future work would be to close the gap between the $\mathcal{O}(n^3/\varepsilon^5)$ upper bound and the $\Omega(n/\log(n))$ lower bound.

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

# A   APPENDIX

## A.1   LOWER BOUND

To complement the upper bound shown in the main paper, we show that the best-known lower bound for the problem in Theorem 1, is $\Omega(n/\log(n))$. This bound is from Canonne et al. (2020).

**Theorem 3** (Theorem 11 in (Canonne et al., 2020))**.** *An absolute constant $\varepsilon_0 < 1$ exists, such as the following holds. Any algorithm that, given a parameter $\varepsilon \in (0, \varepsilon_0]$, and sample access to product distributions $\mathcal{P}, \mathcal{Q}$ over $\{0,1\}^n$, distinguishes between $d_{TV}(\mathcal{P}, \mathcal{Q}) < \varepsilon$ and $d_{TV}(\mathcal{P}, \mathcal{Q}) > 2\varepsilon$, with probability at least 2/3, requires $\Omega(n/\log(n))$ samples. Moreover, the lower bound still holds in the case where $\mathcal{Q}$ is known, and provided as an explicit parameter.*

The lower bound is shown for the case where the tester has access to samples from a product distribution $\mathcal{P}$ and $\mathcal{Q}$(over $\{0,1\}^n$). As observed by Bhattacharyya & Chakraborty (2018), SUBCOND access is no stronger than SAMP when it comes to product distributions. Thus we have the following lower bound:

**Corollary 1.** *Let $\mathcal{S}(\varepsilon_1, \varepsilon_2, \mathcal{P}, \mathcal{Q})$ be any algorithm that has SUBCOND access to distribution $\mathcal{P}$, and explicit knowledge of $\mathcal{Q}$ (defined over $\{0,1\}^n$), and distinguishes between $d_{TV}(\mathcal{P}, \mathcal{Q}) \le \varepsilon_1$ and $d_{TV}(\mathcal{P}, \mathcal{Q}) > \varepsilon_2$ with probability $> 2/3$. Then, $\mathcal{S}$ makes $\Omega(n/\log(n))$ SUBCOND queries.*

## A.2 PROOF OF LEMMA 1

**Lemma 1.** *(Theorem 3.1 in ([Bhattacharyya et al., 2020](#))) For $\delta, \theta \in (0,1)$ and any distribution $\mathcal{P}$, $p_\sigma$ is called a $\theta$-estimate of $\mathcal{P}(\sigma)$, if $1 - \theta \leq p_\sigma / \mathcal{P}(\sigma) \leq 1 + \theta$. Then, for any two distributions $\mathcal{P}$ and $\mathcal{Q}$, given a set of samples $S$ from $\mathcal{P}$, along with the $\theta$-estimates $p_\sigma$ and $q_\sigma$ for each $\sigma \in S$, let $Z$ be the estimate $Z = \frac{1}{|S|} \sum_{i \in S} 1_{q_\sigma > p_\sigma} \left( 1 - \frac{p_\sigma}{q_\sigma} \right)$. If $|S| \geq \frac{(1-\theta) \log(2/\delta)}{8\theta^2}$, then $d_{TV}(\mathcal{P}, \mathcal{Q}) + 4\theta/(1-\theta) \leq Z \leq d_{TV}(\mathcal{P}, \mathcal{Q}) + 4\theta/(1-\theta)$ with probability $1 - \delta$.*

*Proof.* Let $p_\sigma$ and $q_\sigma$ be the $(1 \pm \theta)$ estimates of $\mathcal{P}(\sigma)$ and $\mathcal{Q}(\sigma)$ respectively, then

$$
\begin{aligned}
d_{TV}(\mathcal{P}, \mathcal{Q}) &= \sum_{\sigma \in \{0,1\}^n} \mathbb{1}_{\mathcal{Q}(\sigma) > \mathcal{P}(\sigma)} \left( 1 - \frac{\mathcal{P}(\sigma)}{\mathcal{Q}(\sigma)} \right) \mathcal{Q}(\sigma) \\
&= \sum_{\sigma \in \{0,1\}^n} \mathbb{1}_{q_\sigma > p_\sigma} \left( 1 - \frac{p_\sigma}{q_\sigma} \right) \mathcal{Q}(\sigma) \qquad (1) \\
&\quad + \underbrace{\sum_{\sigma \in \{0,1\}^n} \left( \mathbb{1}_{\mathcal{Q}(\sigma) > \mathcal{P}(\sigma)} \left( 1 - \frac{\mathcal{P}(\sigma)}{\mathcal{Q}(\sigma)} \right) \mathcal{Q}(\sigma) - \mathbb{1}_{q_\sigma > p_\sigma} \left( 1 - \frac{p_\sigma}{q_\sigma} \right) \mathcal{Q}(\sigma) \right)}_{A}
\end{aligned}
$$

The first summand of (1) can be written as $\mathbb{E}_{\sigma \sim \mathcal{Q}} \left[ \mathbb{1}_{q_\sigma > p_\sigma} \left( 1 - \frac{p_\sigma}{q_\sigma} \right) \right]$.

To bound $|A|$, we will split the domain into three sets, $B_1 = \{x : \mathbb{1}_{\mathcal{Q}(\sigma) > \mathcal{P}(\sigma)} = \mathbb{1}_{q_\sigma > p_\sigma}\}$, $B_2 = \{x : \mathbb{1}_{\mathcal{Q}(\sigma) > \mathcal{P}(\sigma)} > \mathbb{1}_{q_\sigma > p_\sigma}\}$ and $B_3 = \{x : \mathbb{1}_{\mathcal{Q}(\sigma) > \mathcal{P}(\sigma)} < \mathbb{1}_{q_\sigma > p_\sigma}\}$.

$$
\begin{aligned}
|A| &= \left| \sum_{\sigma \in \{0,1\}^n} \left( \mathbb{1}_{\mathcal{Q}(\sigma) > \mathcal{P}(\sigma)} \left( 1 - \frac{\mathcal{P}(\sigma)}{\mathcal{Q}(\sigma)} \right) \mathcal{Q}(\sigma) - \mathbb{1}_{q_\sigma > p_\sigma} \left( 1 - \frac{p_\sigma}{q_\sigma} \right) \mathcal{Q}(\sigma) \right) \right| \\
&\leq \sum_{\sigma \in \{0,1\}^n} \left| \left( \mathbb{1}_{\mathcal{Q}(\sigma) > \mathcal{P}(\sigma)} \left( 1 - \frac{\mathcal{P}(\sigma)}{\mathcal{Q}(\sigma)} \right) \mathcal{Q}(\sigma) - \mathbb{1}_{q_\sigma > p_\sigma} \left( 1 - \frac{p_\sigma}{q_\sigma} \right) \mathcal{Q}(\sigma) \right) \right| \\
&= \sum_{\sigma \in B_1} \mathbb{1}_{\mathcal{Q}(\sigma) > \mathcal{P}(\sigma)} \left| \frac{\mathcal{P}(\sigma)}{\mathcal{Q}(\sigma)} - \frac{p_\sigma}{q_\sigma} \right| \mathcal{Q}(\sigma) + \sum_{\sigma \in B_2} \mathbb{1}_{\mathcal{Q}(\sigma) > \mathcal{P}(\sigma)} \left( 1 - \frac{\mathcal{P}(\sigma)}{\mathcal{Q}(\sigma)} \right) \mathcal{Q}(\sigma) \\
&\quad + \sum_{\sigma \in B_3} \mathbb{1}_{q_\sigma > p_\sigma} \left( 1 - \frac{p_\sigma}{q_\sigma} \right) \mathcal{Q}(\sigma)
\end{aligned}
$$

For $\sigma \in B_1$, $\left| \frac{\mathcal{P}(\sigma)}{\mathcal{Q}(\sigma)} - \frac{p_\sigma}{q_\sigma} \right| \leq \frac{2\theta}{1-\theta} \frac{\mathcal{P}(\sigma)}{\mathcal{Q}(\sigma)} \leq \frac{2\theta}{1-\theta}$. For $\sigma \in B_2$, $1 - \frac{\mathcal{P}(\sigma)}{\mathcal{Q}(\sigma)} \leq 1 - \frac{1-\theta}{1+\theta} = \frac{2\theta}{1+\theta}$, and for $\sigma \in B_3$, $1 - \frac{\mathcal{P}(\sigma)}{\mathcal{Q}(\sigma)} \leq 1 - \frac{1-\theta}{1+\theta} = \frac{2\theta}{1+\theta}$. Thus, $|A| \leq \sum_{\sigma \in B_1} \frac{2\theta}{1-\theta} \mathcal{Q}(\sigma) + \sum_{\sigma \in B_2} \frac{2\theta}{1+\theta} \mathcal{Q}(\sigma) + \sum_{\sigma \in B_3} \frac{2\theta}{1+\theta} \mathcal{Q}(\sigma) \leq \frac{2\theta}{1-\theta}$. Plugging the bounds on $|A|$ back into (1), we get

$$
\left| d_{TV}(\mathcal{P}, \mathcal{Q}) - \mathbb{E} \left[ \mathbb{1}_{q_\sigma > p_\sigma} \left( 1 - \frac{p_\sigma}{q_\sigma} \right) \right] \right| \leq \frac{2\theta}{1-\theta}
$$

And hence, $\mathbb{E} \left[ \mathbb{1}_{q_\sigma > p_\sigma} \left( 1 - \frac{p_\sigma}{q_\sigma} \right) \right] - \frac{2\theta}{1-\theta} \leq d_{TV}(\mathcal{P}, \mathcal{Q}) \leq \mathbb{E} \left[ \mathbb{1}_{q_\sigma > p_\sigma} \left( 1 - \frac{p_\sigma}{q_\sigma} \right) \right] + \frac{2\theta}{1-\theta}$.

The distance estimation algorithm draws $|S|$ samples to estimate $\mathbb{E} \left[ \mathbb{1}_{q_\sigma > p_\sigma} \left( 1 - \frac{p_\sigma}{q_\sigma} \right) \right]$. We will use $Z$ to denote the empirical estimate of $\mathbb{E} \left[ \mathbb{1}_{q_\sigma > p_\sigma} \left( 1 - \frac{p_\sigma}{q_\sigma} \right) \right]$. Since each sample $\sigma$ is drawn independently, and $\mathbb{1}_{q_\sigma > p_\sigma} \left( 1 - \frac{p_\sigma}{q_\sigma} \right)$ is bounded in $[0,1]$, we can use the Hoeffding bound as follows,

$$
\Pr \left[ \left| Z - \mathbb{E} \left[ \mathbb{1}_{q_\sigma > p_\sigma} \left( 1 - \frac{p_\sigma}{q_\sigma} \right) \right] \right| \geq \frac{2\theta}{1-\theta} \right] \geq 1 - 2 \exp \left( -2|S| \left( \frac{2\theta}{1-\theta} \right)^2 \right) = 1 - \delta
$$

Hence, with probability at least $1 - \delta$, $Z - \frac{4\theta}{1-\theta} \leq d_{TV}(\mathcal{P}, \mathcal{Q}) \leq \hat{Z} + \frac{4\theta}{1-\theta}$. $\qquad \square$

## B  PROOF OF THEOREM 1

**Theorem 1.** *Given two distributions $\mathcal{P}$ and $\mathcal{Q}$ over $\{0,1\}^n$, and the parameters $\varepsilon \in (0,1)$, and $\delta \in (0,1)$, the algorithm $\mathsf{DistEstimate}(\mathcal{P}, \mathcal{Q}, \varepsilon, \delta)$ returns $\kappa$ such that*

$$\Pr[d_{TV}(\mathcal{P}, \mathcal{Q}) - \varepsilon \le \kappa \le d_{TV}(\mathcal{P}, \mathcal{Q}) + \varepsilon] \ge 1 - \delta$$

$\mathsf{DistEstimate}$ *makes $\tilde{\mathcal{O}}\left(n^3 \log(1/\delta)/\varepsilon^5\right)$ queries to the $\mathsf{SUBCOND}$ oracle.*

*Proof.* From Thm 2 we know that each call to $\mathsf{DistEstimateCore}$ returns an estimate $\kappa$ that is within $\pm\varepsilon$ of the actual $d_{TV}(\mathcal{P}, \mathcal{Q})$ with probability at least $5/6$. We use the standard median trick to boost this probability to $1 - \delta$ at the cost of $K = 48 \log(1/\delta)$ independent repetitions (Line 1 of Alg. 1). Now we can analyze the overall query complexity of the algorithm. There are atmost $\tilde{O}\left(n^3/\varepsilon^5\right)$ $\mathsf{SUBCOND}$ queries in each call to $\mathsf{DistEstimateCore}$, and since $\mathsf{DistEstimateCore}$ is called $O(\log(1/\delta))$ times, the total number of queries is in $\tilde{O}\left(n^3 \log(1/\delta)/\varepsilon^5\right)$. □

## C  PROOF OF LEMMA 2

**Lemma 2.** *For $\theta \in (0, 1/2)$, distribution $\mathcal{D}$, and its $\theta$-tamed version $\mathcal{D}'$, we have $d_{TV}(\mathcal{D}, \mathcal{D}') \le \theta n$.*

*Proof.* Recall the definition of subcube $S_\rho = \{w \in \{0,1\}^n : w_{\le |\rho|} = \rho\}$ and of the distribution $\mathcal{D}_\rho$ (defined in Lemma 6). We again state the definition of $\mathcal{D}_\rho$ here for convenience. For any distribution $\mathcal{D}$, string $\rho$ (with $1 \le |\rho| \le n$) and $\omega \in \{0,1\}^{n-|\rho|}$, the distribution $\mathcal{D}_\rho$ denotes the marginal distribution of $\mathsf{SUBCOND}(\mathcal{D}, \rho)$ in the remaining dimensions, i.e. for any $\omega \in \{0,1\}^{n-|\rho|}$, $\mathcal{D}_\rho(\omega) = \Pr_{w \sim \mathsf{SUBCOND}(\mathcal{D}, \rho)}[w = \rho\omega]$.

Consider the induction hypothesis that $d_{TV}(\mathcal{D}, \mathcal{D}') \le \theta i$ if $\mathcal{D}$ is supported on $\{0,1\}^i$. To verify the hypothesis for $i = 1$, wlog assume that $\mathcal{D}(0) \le \mathcal{D}(1)$, then $d_{TV}(\mathcal{D}, \mathcal{D}') = \mathcal{D}(1) - \mathcal{D}'(1) = 2\theta\mathcal{D}(1) - \theta \le \theta$. Assume the hypothesis holds for all $i \in [n-1]$. Now, we show the hypothesis is true for $i = n$.

Consider a distribution $\mathcal{D}$ over $\{0,1\}^n$ and its $\theta$-tamed counterpart $\mathcal{D}'$, then:

$$d_{TV}(\mathcal{D}, \mathcal{D}') = \frac{1}{2} \sum_{\sigma \in \{0,1\}^n} |\mathcal{D}(\sigma) - \mathcal{D}'(\sigma)| = \frac{1}{2} \sum_{\rho \in \{0,1\}} \sum_{\omega \in \{0,1\}^{n-1}} |\mathcal{D}(\rho\omega) - \mathcal{D}'(\rho\omega)|$$

$$= \frac{1}{2} \sum_{\rho \in \{0,1\}} \sum_{\omega \in \{0,1\}^{n-1}} |\mathcal{D}(S_\rho)\mathcal{D}_\rho(\omega) - \mathcal{D}'(S_\rho)\mathcal{D}'_\rho(\omega)|$$

$$= \frac{1}{2} \sum_{\rho \in \{0,1\}} \sum_{\omega \in \{0,1\}^{n-1}} |\mathcal{D}(S_\rho)\mathcal{D}_\rho(\omega) - \mathcal{D}(S_\rho)\mathcal{D}'_\rho(\omega) + \mathcal{D}(S_\rho)\mathcal{D}'_\rho(\omega) - \mathcal{D}'(S_\rho)\mathcal{D}'_\rho(\omega)|$$

$$\le \frac{1}{2} \sum_{\rho \in \{0,1\}} \sum_{\omega \in \{0,1\}^{n-1}} |\mathcal{D}(S_\rho)\mathcal{D}_\rho(\omega) - \mathcal{D}(S_\rho)\mathcal{D}'_\rho(\omega)| + |\mathcal{D}(S_\rho)\mathcal{D}'_\rho(\omega) - \mathcal{D}'(S_\rho)\mathcal{D}'_\rho(\omega)|$$

$$= \frac{1}{2} \sum_{\rho \in \{0,1\}} \sum_{\omega \in \{0,1\}^{n-1}} \mathcal{D}(S_\rho)|\mathcal{D}_\rho(\omega) - \mathcal{D}'_\rho(\omega)| + \mathcal{D}'_\rho(\omega)|\mathcal{D}'(S_\rho) - \mathcal{D}(S_\rho)|$$

$$= \frac{1}{2} \sum_{\rho \in \{0,1\}} \left(\mathcal{D}(S_\rho)2d_{TV}(\mathcal{D}_\rho, \mathcal{D}'_\rho)\right) + \frac{1}{2} \sum_{\rho \in \{0,1\}} |\mathcal{D}'(S_\rho) - \mathcal{D}(S_\rho)|$$

$$\le \sum_{\rho \in \{0,1\}} \left(\mathcal{D}(S_\rho)\theta(n-1)\right) + \theta = \theta n$$

We use $|a + b| \le |a| + |b|$ in the first inequality. In the second, we use the induction hypothesis to bound the first summand, and for the second, we observe that for $c \in \{0,1\}$, $|\mathcal{D}'(c) - \mathcal{D}(c)| \le \theta$. □

## D  PROOF OF LEMMA 4

**Lemma 4.** *For $j \in [n]$, the random variable $x_j$ (in Algorithm 4) is distributed as $\mathsf{NB}(k, \mathcal{D}_{\sigma_{<j}}^m(\sigma_j))$*

*Proof.* Fix any $j \in [n]$. In Algorithm 4, the r.v $\alpha_j$ takes the value $\sigma_j$ with probability $\mathcal{D}_{\sigma_{<j}}^m(\sigma_j)$. Note that while the value of $x_j$ increments by one in every iteration of the loop (lines 5-8), the value of $t$ increases by one only when $\alpha_j = \sigma_j$. Since the loop (lines 5-8) runs until the value of $t$ is $k$, the distribution of $x_j$ is $\mathsf{NB}(k, \mathcal{D}_{\sigma_{<j}}^m(\sigma_j))$. $\square$

## E  PROOF OF LEMMA 6

**Lemma 6.** *For a distribution $\mathcal{D}$ defined over $\{0,1\}^n$, $\varepsilon \in (0,1)$,*
$$\mathbb{E}_{\sigma \sim \mathcal{D}} \left[ \mathbb{E} \left[ \mathsf{QC}(\mathsf{SubToEval}'(\mathcal{D}, \varepsilon, \sigma)) \right] \right] = \lceil 8n^2 \varepsilon^{-2} \rceil$$
*where the inner expectation is over the internal randomness of $\mathsf{SubToEval}'$*

*Proof.* The number of SUBCOND calls made by $\mathsf{SubToEval}'$ in the $j^{th}$ iteration is captured by $x_j$. The total query complexity is the sum of $x_j$ over the $n$ iterations,
$$\mathsf{QC}(\mathsf{SubToEval}'(\mathcal{D}, \varepsilon, \sigma)) = \sum_{j=1}^{n} x_j$$

Recall from Lemma 4 that $x_j$ is modeled as a random variable coming from the negative binomial distribution $\mathsf{NB}(k, \mathcal{D}_{\sigma_{<j}}^m(\sigma_j))$. Hence, we have $\mathbb{E}[x_j] = k/\mathcal{D}_{\sigma_{<j}}^m(\sigma_j)$. Then, the expected sample complexity is given as,
$$\mathbb{E} \left[ \mathsf{QC}(\mathsf{SubToEval}'(\mathcal{D}, \varepsilon, \sigma)) \right] = \mathbb{E} \left[ \sum_{j=1}^{n} x_j \right] = \sum_{j=1}^{n} \mathbb{E}[x_j] = \sum_{j=1}^{n} \frac{k}{\mathcal{D}_{\sigma_{<j}}^m(\sigma_j)} = \lceil 4n\varepsilon^{-2} \rceil \sum_{j=1}^{n} \frac{1}{\mathcal{D}_{\sigma_{<j}}^m(\sigma_j)} \tag{2}$$

In the last equality, we substitute $k$ from Line 1 of $\mathsf{SubToEval}'$.

We now introduce some notation that will be used throughout this lemma. For any distribution $\mathcal{D}$, and strings $\rho$ (with $1 \le |\rho| \le n$) and $\omega \in \{0,1\}^{n-|\rho|}$, the distribution $\mathcal{D}_\rho$ denotes the marginal distribution of $\mathsf{SUBCOND}(\mathcal{D}, \rho)$ in the remaining dimensions, i.e. for any $\omega \in \{0,1\}^{n-|\rho|}$, $\mathcal{D}_\rho(\omega) = \mathrm{Pr}_{\pi \sim \mathsf{SUBCOND}(\mathcal{D}, \rho)}[\pi = \rho\omega]$ Also, recall the definition of subcube $S_\rho = \{\pi \in \{0,1\}^n : \pi_{\le |\rho|} = \rho\}$. Now, we will determine the expectation of the expected sample complexity over the distribution $\mathcal{D}$,

$$\mathbb{E}_{\sigma \sim \mathcal{D}} \left[ \mathbb{E} \left[ \mathsf{QC}(\mathsf{SubToEval}'(\mathcal{D}, \varepsilon, \sigma)) \right] \right] = \mathop{\mathbb{E}}_{\sigma \sim \mathcal{D}} \left[ \lceil 4n\varepsilon^{-2} \rceil \sum_{j=1}^{n} \frac{1}{\mathcal{D}_{\sigma_{<j}}^m(\sigma_j)} \right]$$

$$= \lceil 4n\varepsilon^{-2} \rceil \sum_{j=1}^{n} \mathop{\mathbb{E}}_{\sigma \sim \mathcal{D}} \left[ \frac{1}{\mathcal{D}_{\sigma_{<j}}^m(\sigma_j)} \right] = \lceil 4n\varepsilon^{-2} \rceil \sum_{j=1}^{n} \sum_{\sigma \in \{0,1\}^n} \frac{\mathcal{D}(\sigma)}{\mathcal{D}_{\sigma_{<j}}^m(\sigma_j)}$$

$$= \lceil 4n\varepsilon^{-2} \rceil \sum_{j=1}^{n} \sum_{\rho \in \{0,1\}^{j-1}} \sum_{\omega \in \{0,1\}^{n-j+1}} \frac{\mathcal{D}(\rho\omega)}{\mathcal{D}_\rho^m(\omega_1)}$$

$$= \lceil 4n\varepsilon^{-2} \rceil \sum_{j=1}^{n} \sum_{\rho \in \{0,1\}^{j-1}} \mathcal{D}(S_\rho) \sum_{\omega \in \{0,1\}^{n-j+1}} \frac{\mathcal{D}_\rho(\omega)}{\mathcal{D}_\rho^m(\omega_1)} \quad \text{(Since } \mathcal{D}(\rho\omega) = \mathcal{D}(S_\rho)\mathcal{D}_\rho(\omega))$$

$$= \lceil 4n\varepsilon^{-2} \rceil \sum_{j=1}^{n} \sum_{\rho \in \{0,1\}^{j-1}} \mathcal{D}(S_\rho) \sum_{\omega_1 \in \{0,1\}} \frac{\mathcal{D}_\rho^m(\omega_1)}{\mathcal{D}_\rho^m(\omega_1)}$$

$$= \lceil 4n\varepsilon^{-2} \rceil \sum_{j=1}^{n} \sum_{\rho \in \{0,1\}^{j-1}} 2\mathcal{D}(S_\rho)$$

$$= \lceil 4n\varepsilon^{-2} \rceil \sum_{j=1}^{n} 2 = \lceil 8n^2\varepsilon^{-2} \rceil$$

$\square$

## F   ADDITIONAL EXPERIMENTS

### F.1   CALIBRATION

In this experiment, we compare our algorithm against the naive learning-based distance estimator, which we consider as the ground truth. We run our estimator with $\varepsilon = 0.3$, which indicates that the result is expected to be within 0.3 of the ground truth. We run the study on 10 benchmarks. In the following plots, we present the output of DistEstimateCore against the ground truth. We find that in all cases, our estimate was within the tolerance used for the estimate.

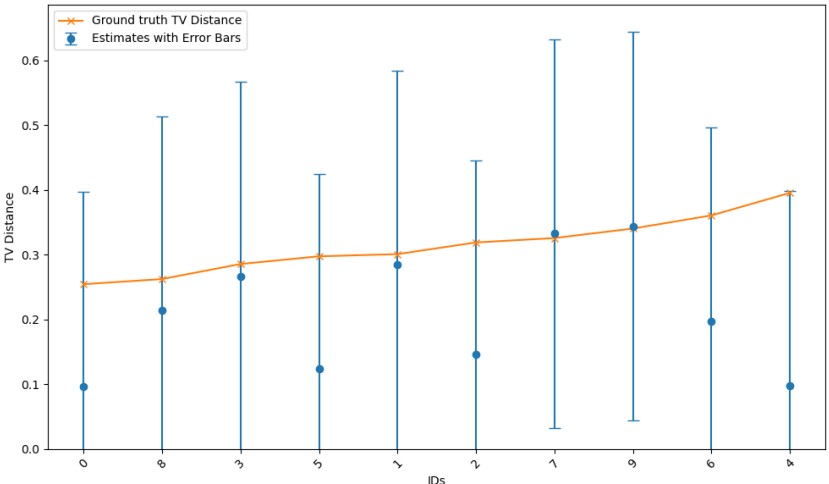

Figure 2: In this figure, we plot the distance of the distribution generated by CMSGen. The blue lines are the range of the error, which we have set to $\varepsilon = 0.3$

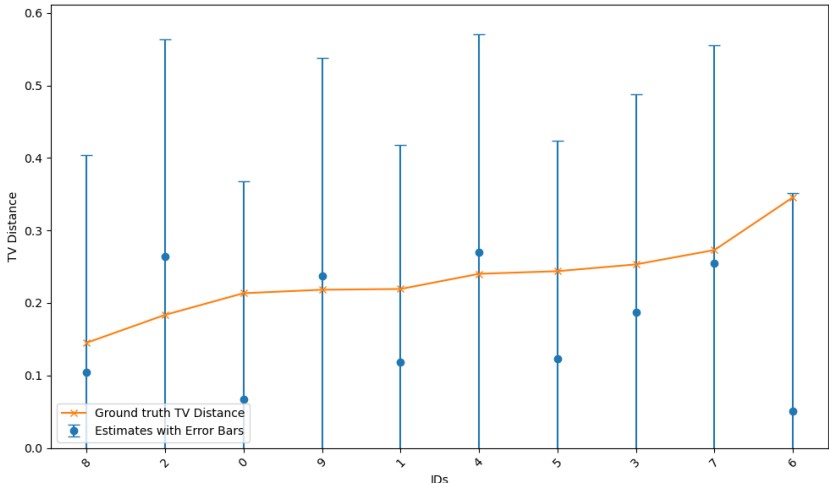

Figure 3: In this figure, we plot the distance of the distribution generated by STS. The blue lines are the range of the error, which we have set to $\varepsilon = 0.3$.

