# OpenReview forum: "Distance Estimation for High-Dimensional Distributions"
_ICLR.cc/2024/Conference — Submitted to ICLR 2024_

### Official Review · Reviewer_F6ny · 2023-10-18

**Soundness:** 2 fair
**Presentation:** 2 fair
**Contribution:** 3 good
**Rating:** 8
**Confidence:** 3

**Summary:**

This paper studies the problem of estimating the distances between two distributions $P$ and $Q$ over the $n$-dimensional Boolean hypercube. Since the problem is known to require $\Omega(2^n/n)$ samples from $P$ and $Q$ to estimate the distance up to an additive constant, this paper considers the subcube conditioning query model, where each query can be made from $P$ and $Q$ conditioned on a specified subcube of the domain. This paper gives an algorithm for estimating the distance to additive $\varepsilon$ using $\tilde{O}(n^3/\varepsilon^5)$ queries as well as a lower bound showing that $\Omega(n/\log n)$ queries are necessary in the subcube conditioning model for constant $\varepsilon$.

**Strengths:**

+ To the best of my knowledge, the problem of distance estimation has not previously been studied in the subcube conditioning model.
+ The polynomial query complexity in the subcube conditioning model is an exponential improvement over the known $\Omega(2^n/n)$ query complexity lower bound in the standard model.

**Weaknesses:**

- Although there is good intuition for parts of the main result, I think the paper could benefit from additional algorithmic/analytic intuition rather than the full formal proofs. For example, I would have liked to see more details about how the tamed distribution could be accessed using subcube conditioning queries. I would have also liked to see intuition on where the $n^3$ and $1/\varepsilon^5$ factors come from.
- It is not clear to me that the applications of distance estimation with subcube conditioning queries is well-suited for the particular learning theory community at ICLR.
- Although I do not think experiments are necessary for a paper with solid theoretical foundations, I think the experimental section of this paper is a bit unclear (see questions below).

In summary, my concern is that the current presentation of the paper may not supply sufficient context for a theoretical audience but the applications/experiments are also not extensive enough for an applied audience. Hopefully the authors can respond to this concern in the discussion phase.

**Questions:**

1) How is the distance estimation problem applied to constrained samplers?
2) How was the scalable benchmark generated and how are the real-world circuits represented by the Boolean formulas?
3) How does the tradeoff between sample complexity and TVD look like for these datasets?
4) What are additional applications of distance estimation with subcube conditioning queries?

EDIT (Post-rebuttal): Thanks for answering my questions. I think my main concerns about the intended audience at ICLR have been addressed by additional discussions on 1) applications of subcube conditioning queries, though perhaps slightly less mainstream and 2) intuition on the main algorithm and proof. Although I think this paper is a good theoretical work by itself, I see that additional experiments have also been included in the discussion phase. Thus I have adjusted my score accordingly.

---

> ### Author Response · Authors · 2023-11-23
>
> We have reorganized the exposition of the proof to make it more intuitive. Additionally, we have fleshed out the part about taming to make the role of SUBCOND clearer. Further, we have also added an informal description of SubToEval with the aim of making it clear how we arrived at $n^3$ and $\varepsilon^{-5}$.
>
>
> Q1.How is the distance estimation problem applied to constrained samplers?
> Ans. Constrained samplers are required to sample approximately from a specified distribution, and their performance is judged by the distance of their output distribution to the specification. One application of our distance estimation algorithm is a method to test the performance of heuristic constrained samplers by measuring their distance to a distribution that is guaranteed to be exactly as specified.
>
>
> Q2. How was the scalable benchmark generated, and how are the real-world circuits represented by the Boolean formulas?
> Ans. As mentioned in the experimental evaluation, the scalable benchmarks are random 3-cnfs of varying numbers of variables. On the other hand, the real-world circuits are already available in CNF and were obtained from a set of publicly available benchmarks arising from sampling and counting tasks (https://zenodo.org/records/3793090). We have added details regarding the benchmarks to the revised version of the paper.
>
>
> Q3. How does the tradeoff between sample complexity and TVD look like for these datasets?
> Ans. We will analyze the sample complexity with respect to TVD and present it in a later revision.
>
>
> Q4. What are the additional applications of distance estimation with subcube conditioning queries?
> Ans. Distance estimation with subcube conditioning has applications in other fields, such as databases and probabilistic circuits, as subcube conditional queries are natural to these domains.

---

> > ### Comment · Reviewer_F6ny · 2023-12-04
> >
> > Thanks for answering my questions. I think my main concerns about the intended audience at ICLR have been addressed by additional discussions on 1) applications of subcube conditioning queries, though perhaps slightly less mainstream and 2) intuition on the main algorithm and proof. Although I think this paper is a good theoretical work by itself, I see that additional experiments have also been included in the discussion phase. Thus I have adjusted my score accordingly.

---

### Official Review · Reviewer_SPku · 2023-10-22

**Soundness:** 1 poor
**Presentation:** 1 poor
**Contribution:** 1 poor
**Rating:** 1
**Confidence:** 4

**Summary:**

This paper claims itself presents the first polynomial sample distance estimator in the conditional sampling model. This paper provides detailed proofs and simple experiments for this paper.

**Strengths:**

This paper presents the first polynomial sample distance estimator in the conditional sampling model. I roughly check the proof, and they are OK to me at this time. Consequently, in theory aspect, I am satisfied with authors' contributions.

**Weaknesses:**

Although I am satisfied with authors' theoretical contributions, I am "extremely' unsatisfied with experiment part of this paper. Figure 1 and Table 1 just do not make any sense to me. If authors could improve that part, I will largely raise my rating.

1. Please conduct 'enough' experiments to justify your theorem. This can be plots of repeated experiments showing the relationships between $n$, $\delta$, and $\epsilon$, which is widely used in this area.

2. Dimensions in your experiments are not high enough, and $\epsilon$ is too large.

3. If possible, provide a comparison of your method with other existing methods, which could further demonstrate the significance of your method.

**Questions:**

Is that possible to provide a high probability bound with respect to the number of queries? If not, could your provide the technical hardness?

---

> ### Author Response · Authors · 2023-11-23
>
> Q) Is it possible to provide a high probability bound with respect to the number of queries? If not, could you provide the technical hardness?
> A) Yes, we can show a high-probability bound with the help of the median trick. We have updated the proof in the revised version to show the mentioned bound.
>
>
>
> Regarding Experiments:
> There has been no prior work that has been able to scale to even ~40 dimensions for the case of constrained sampling, as constrained samplers tend to be slow. We would like to point out that the previous estimators would have required significantly more time and samples due to their exponential(in n) sample complexity.
>
> We were not able to perform experiments with larger dimensions due to the lack of computing resources. Currently, every instance requires around 4*10^5 s = 5 days of computer time.
>
> As mentioned in the paper previously, there were no distance estimators that drew only poly(n) samples. In a later revision, we will add a discussion of the previous exponential sample algorithm to our related work section to place our work better.

---

### Official Review · Reviewer_4aJb · 2023-10-31

**Soundness:** 4 excellent
**Presentation:** 4 excellent
**Contribution:** 3 good
**Rating:** 6
**Confidence:** 4

**Summary:**

It is known that estimating TV distance between two discrete probability distributions over $\{0,1\}^n$ has a sample complexity lower bound of $\Omega(2^n/n)$. In this work, the authors focus on a more powerful model, namely SUBCOND, which takes a prefix as input and returns the full string according to the conditional distribution. With this model, the authors design DistEstimate algorithm that makes $\tilde{O}(n^3\log(1/\delta)/\varepsilon^5)$ calls to SUBCOND and returns an estimate of the TV distance with margin of error $\varepsilon$ with probability $1-\delta$.

**Strengths:**

- The authors provide clear motivation in terms of why a stronger model is needed for distance estimation.
- The explanation of the algorithm is clear.
- The proof of the main theorem is mostly self-contained. The proofs, from a quick read, seem sound. The technique of computing expectation of number of queries using properties of negative binomial distributions is really nice.
- A direction for future research is also discussed.

**Weaknesses:**

- There needs to be an example in the introduction that describes scenarios in which the SUBCOND query is available. Personally, I have been very curious in such scenarios until I read the Application section.
- In the experiment, $\varepsilon=0.5$ is too large considering the scale of the TV distances. Also, there should be error bars in Figure 1, at least for small numbers of dimensions in order to demonstrate the stability of the algorithm.

**Questions:**

- Algorithm 1 returns the average of  $1-p_i-q_i$. Should the definition of $Z$ in Lemma 1 be divided by $m$ as well?
- I suggest the authors add a simulation to confirm that DistEstimate attains $\varepsilon$-estimation error with probability at least $1-\delta$.

Minor comments:
- The notations for the probability distributions are not consistent. Some times they are $P,Q$, the other times they are $\mathcal{P},\mathcal{Q}$. Please check throughout the paper.

---

> ### Author Response · Authors · 2023-11-23
>
> We thank the reviewer for their careful reading and helpful suggestions.  Z was indeed missing a factor of m, which we have corrected in the updated version.
>
>
> Q) There needs to be an example in the introduction that describes scenarios in which the SUBCOND query is available. Personally, I have been very curious in such scenarios until I read the Application section..
>
>
> A) For our experiments we focus on samplers that sample satisfying assignments of a CNF, and \subcond is particularly well suited for this problem, as a CNF conditioned on a subcube is also a CNF. This enables us to recursively solve the problem as the subproblems are of the same type. As suggested, we have added the example from the application section to the introduction of our paper.

---

### Official Review · Reviewer_YTK5 · 2023-11-07

**Soundness:** 4 excellent
**Presentation:** 3 good
**Contribution:** 4 excellent
**Rating:** 8
**Confidence:** 4

**Summary:**

This paper considers the problem of estimating the total variation distance between distributions $P$ and $Q$ on $\{0,1\}^n$. This problem is known to require $\exp(n)$ samples if we only get access to queries. This paper considers the SUBCOND model, where you are allowed to sample from a distribution conditioned on a certain prefix. It proves that one can estimate the TV distance with query complexity $poly(n, 1/\epsilon)$. I found this statement surprising (although I am not familiar with the literature in the area which is vast).   In terms of techniques, the paper seems to make use of prior results which adding some new ingredients of its own. I think this is a nice contribution that ought to be accepted.

**Strengths:**

1. The problem of TV estimation is important. The existing lower bounds are very strong, which motivates looking for other models where it is easier to estimate. The SUBCOND model is fairly natural for the Boolean hypercube setting.

2. I found the result in itself surprising, though I am not an area expert. After understanding the paper better, maybe I am a little less surprised, but I still think its a great result.

3. Technically, the paper is sound, and above the bar for acceptance. There seem to be two novel ideas, the simpler one is to  "tame" the distribution so that there is some non-trivial probability on both the subcubes $x_i = 0$ and $x_i =1$. This is achieved essentially by adding some noise to the distribution. The second is using the conditioning oracle to get a multiplicative approximation to the probability of string, using the formula $\Pr[x =a] = \Pr[x_1 =a_1]\cdot \Pr[x_2 =a_2|x_1 =a_1] \cdots$.  It is simple and elegant. They plug this into a previous result that lets you estimate the total variation distance given good enough estimates of the importance weights.

**Weaknesses:**

1. The title promises too much compared to what the paper delivers. When most people think of high dimensional distributions, they don't have the Boolean hypercube in mind as the domain, they would think of $\mathbb{R}^n$. I would suggest adding "discrete distributions" to the title, and possibly mentioning the need for conditional samples.

2. On a related note, the SUBCOND model is very natural in the discrete setting (for product domains). I wonder to what extent these techniques can extend to other domains, and what a reasonable analog of this model might be.

**Questions:**

- It appears that your results allow you to estimate the importance weights  $Q(x)/P(x)$ for $x \sim Q$. If so, this might be useful for estimating various other divergences such as KL and Renyi.

---

> ### Author Response · Authors · 2023-11-22
>
> As suggested, we have changed the title to “Distance Estimation for High-Dimensional Discrete Distributions” to reflect our contributions more accurately.
>
>
> Our contribution extends to the hypergrid setting $[d_1] \times [d_2] \times \ldots [d_n]$, where $d_i$ is any discrete set, and we will add this to a later revision. We note that in practice, reals are represented as discrete floats, making SUBCOND potentially useful in some settings with real dimensions.
>
>
> We thank the reviewer for their useful suggestions regarding possible extensions of our technique to other divergences.

---

### Author Response · Authors · 2023-11-23

Dear Reviewers,

Thank you for the initial reviews. We have made changes to the paper based on the reviews and tried to alleviate some of the major concerns in the rebuttal and in the paper.

Specifically, we have made the following major edits:

1. We have rewritten the main algorithm and its proof so that there is a high probability bound on the sample complexity.
2. We have reorganized the exposition of the proof and added an informal description to make it easier to understand.
3. We have added a new experiment (appendix Section F) to show the calibration between our estimator and the ground truth.

–
Authors

---

### Meta-Review · Area_Chair_5Unx · 2023-12-22

**Metareview:**

The paper studies the problem of estimating TV distance between two distributions over n-dimensional Boolean hypercubes. The authors propose a new estimator that achieves a polynomial (in n) query complexity when one can obtain conditional samples. Previous known lower bounds in a non-conditional setting are exponential in the dimension n. The paper also provides a lower bound on the query complexity of a conditional estimator, which is a direct consequence of previous work. The authors then run their algorithm, measuring the runtime, on a few different datasets. The improvements in the sample complexity bound seem significant, though a large gap between the lower and upper bounds remains. The paper is fairly well-written and overall the results are presented quite clearly. That said, the paper could be made more accessible by adding additional intuition and a high level overview of the proofs.

**Justification For Why Not Higher Score:**

The paper tackles a statistical problem. As noted by the reviewers, it seems like sample complexity bounds for distance estimation with subcube conditioning queries may not be well-suited for the ICLR community. There are some potential connections to ML literature (e.g., GANs, diffusion models, etc.), but the authors made no effort to connect their work to ML. In the paper, the algorithm was applied to really low dimensional settings, which also makes me question to what extent it could be used for higher dimensional distributions common in ML. Finally, the domain is quite restricted (distributions are over n-dimensional Boolean hypercubes).

**Justification For Why Not Lower Score:**

N/A

---

### Decision · Program_Chairs · 2024-01-16

Reject